# Dynamic Foam Characteristics during Cultivation of *Arthrospira platensis*

**DOI:** 10.3390/bioengineering9060257

**Published:** 2022-06-16

**Authors:** Ameer Ali Kubar, Amjad Ali, Santosh Kumar, Shuhao Huo, Muhammad Wajid Ullah, Khulood Fahad Saud Alabbosh, Muhammad Ikram, Jun Cheng

**Affiliations:** 1School of Food and Biological Engineering, Jiangsu University, Zhenjiang 212013, China; ameerali.kubar@yahoo.com; 2State Key Laboratory of Clean Energy Utilization, Zhejiang University, Hangzhou 310027, China; santoshmalhi@zju.edu.cn; 3Research School of Polymeric Materials, School of Materials Science and Engineering, Jiangsu University, Zhenjiang 212013, China; amjadali@zju.edu.cn; 4Biofuels Institute, School of the Environment and Safety Engineering, Jiangsu University, Zhenjiang 212013, China; wajid_kundi@ujs.edu.cn; 5Department of Biology, College of Science, University of Hail, Hail 55476, Saudi Arabia; k.alabosh@uoh.edu.sa; 6Department of Pharmacy, COMSATS University Islamabad, Abbottabad Campus, Abbottabad 22060, Pakistan; ikram@cuiatd.edu.pk

**Keywords:** foam, stability, flow rate, photobioreactor, carbon dioxide, temperature

## Abstract

This study is aimed at understanding the serious foaming problems during microalgal cultivation in industrial raceway ponds by studying the dynamic foam properties in *Arthrospira platensis* cultivation. *A. platensis* was cultivated in a 4 L bowl bioreactor for 4 days, during which the foam height above the algal solution increased from 0 to 30 mm with a bubble diameter of 1.8 mm, and biomass yield reached 1.5 g/L. The algal solution surface tension decreased from 55 to 45 mN/m, which favored the adsorption of microalgae on the bubble to generate more stable foams. This resulted in increased foam stability (FS) from 1 to 10 s, foam capacity (FC) from 0.3 to 1.2, foam expansion (FE) from 15 to 43, and foam maximum density (FMD) from 0.02 to 0.07. These results show a decrease in CO_2_ flow rate and operation temperature when using the Foamscan instrument, which minimized the foaming phenomenon in algal solutions to a significantly lower and acceptable level.

## 1. Introduction

Microalgae have been the subject of growing industrial research for years due to their greater photosynthetic efficiency in utilizing CO_2_, sunlight, and inorganic nutrients, which in turn contribute to reducing the effects of global warming [1,2]. Many studies have indicated that microalgae are an important source of high-value products and a strong candidate for sustainable and eco-friendly energy sources worldwide; this brings the potential to meet the need of rapidly growing economies, particularly in developing countries, by offering fewer adverse atmospheric effects [3,4,5,6,7]. The microalgal raceway ponds are usually 15 to 25 cm deep, supplied with a source of CO_2,_ equipped with a paddle wheel, and have guided barriers in the flow channel to ensure homogeneity. These raceway ponds are generally lit by sunlight under controlled temperature and have a ready source of water; this is currently the most industrially marketable technology because of its lower costs compared to the closed bioreactors [8,9,10,11,12,13].

Foaming is desirable for a number of global industries, such as in the food industry, where foam is of great importance for basic food textures owing to its lightness and large specific surface area [14,15,16]. Several engineering processes, however, are directly affected by the unwanted foams, such as in metal industries, wastewater treatment plants, and anaerobic digesters. Foams can obstruct gas transport and render the process inefficient, thus significantly increasing cost [17,18,19,20]. Foam becomes problematic during algal cultivation when it is formed to a level that can hinder the regular process tasks. This phenomenon has remained unnoticed and been ignored for decades. Foams may last for short times (up to hours) on solution surfaces during large-scale algal cultivation in an open pond, as shown in Figure 1. This results in adverse effects, such as a decrease in growth, loss of volume, lower light penetration (since phototrophic microalgal cultivation strongly depends on light energy), pond structure failure, and difficult manual cleaning of the reactor. Such impacts can lead to economic losses of varying magnitudes. Numerous studies have provided advances in structural engineering in reactor design for attaining maximum algal biomass yield; however, the formation of foam during algal cultivation in photo-bioreactors is still unreported in the existing literature [21,22,23,24].

To understand the serious foaming problems that present during microalgal cultivation in industrial raceway ponds, the dynamic foam properties from *A. platensis* cultivation were characterized in this study to clarify the processes in foam formation and foam evolution. The reduced surface tension of algal cultures, combined with ionic sulfate and phosphate surfactant adsorption on bubble walls, was found to generate more stable foams. This resulted in an increase in foam stability, foam capacity, foam expansion, and foam maximum density.

## 2. Materials and Methods

### 2.1. Instrumental Setup

The main instrument used in this study was a Foamscan system (Teclis Instruments, Civrieux-d’Azergues, France). A schematic diagram of the Foamscan instrument is shown in Figure 2. This system is connected with software for precise determination of foam volume, liquid fraction and volume, and controlling the gas flow rate, stirring speed, and measuring the bubble size and distribution. Foamscan optically measures the foaming capabilities by providing information on stability, drainage rate, and bubble size distribution [25]. Additionally, the foam column is layered with a temperature sensor inside the tube. In the present study, the Foamscan system was used to determine the foam stability (FS), foam capacity (FC), foam expansion (FE), and foam maximum density (FMD).

### 2.2. Foam Generation and Stability Analysis

For foam generation, a 50 mL sample solution was injected through an inlet into the foam column, equipped with a temperature sensor inside the tube. The experimental parameters were set by using the connected software. For all experiments, the final foam volume limit was set at 70 mL. The foam was generated by sparging CO_2_ gas into the injected sample solution in the foam column, and its volume was controlled with the help of the connected software. Gas sparging was automatically stopped once the foam volume reached the preset final limit, and tests were performed to separately investigate the effects of CO_2_ and temperature. For CO_2_ flow rate tests, the temperature was fixed at 30 °C, whereas CO_2_ aeration was varied according to the testing parameters (200, 250, 300, 350, and 400 mL/min). The temperature effect was determined at varying temperatures (20, 25, 30, 35, and 40 °C), whereas the CO_2_ aeration rate was fixed at 300 mL/min. The foam stability, also known as the half-life, was determined in terms of FE, FC, and FMD, as described below:

FE was determined as the ratio of the total foam volume to the liquid volume within the bubbles after foam generation is completed by using Equation (1).
(1)$$\;\mathrm{FE}\;=\frac{{\mathrm{Vf}}_{\mathrm{foam}}}{{\mathrm{Vi}}_{\mathrm{liq}}-{\;\mathrm{Vf}}_{\mathrm{liq}}}$$
where ${\mathrm{Vf}}_{\mathrm{foam}}\;$(mL) is the total foam volume after the completion of the foaming process, ${\mathrm{Vi}}_{\mathrm{liq}}$ (mL) is the liquid volume at the initial state, and ${\mathrm{Vf}}_{\mathrm{liq}}\;$(mL) is the volume of liquid remaining after the completion of the foaming process.

FC was determined as the ratio of the total foam volume to the gas volume after foam generation is completed by using Equation (2) [26].
(2)$$\;\mathrm{FC}\;=\frac{{\mathrm{Vf}}_{\mathrm{foam}}}{{\mathrm{Vf}}_{\mathrm{gas}}}$$

FMD was determined as the ratio of the difference between the initial liquid volume and the final liquid volume with the final foam volume.

### 2.3. Surface Tension

The surface tension (mN/m) of the solution was measured by using a digital surface pressure device via the du Noüy ring method. Briefly, the fluid samples were added to a container with an automated adjustable height. After each reading, the ring was thoroughly washed with double distilled water and then heated in a flame. Measurements were performed three times for reproducibility and accuracy.

### 2.4. Cultivation Conditions

The *A*. *platensis* strain was cultivated with 4 L Zarrouk’s medium in a bowl photobioreactor aerated with the 15% CO_2,_ and the flow rate was controlled at 300 mL/min using a mass flow meter (Sevenstar CS200, Beijing, China). The experiment was conducted in an artificial climate greenhouse with an indoor temperature of 30 ± 2 °C, and the light intensity was kept at 8000 ± 200 lux. Biomass was measured using optical density with algal solution (10 mL) using a WFJ 7200 visible spectrophotometer at the wavelength of 560 nm with deionized water as blank. Error bars shown were found with Excel’s standard deviation (SD) function. The pH value throughout the study was determined using an FE20 laboratory pH meter. The data for the dry weight curve of the *A. platensis* was obtained by (y = 0.51x − 0.034, where y refers to dry weight, and x refers to OD). Each measurement during all experiments was performed twice a day at 9:00 and 21:00 to ensure reproductively and accuracy of the results, and each data throughout the figures is a mean of three data with an error bar to represent uncertainty in the measurements [26].

### 2.5. Microscopic Observation

The bubble diameter of foam was determined, as described previously [22], by using a Nikon inverted fluorescence microscope (Nikon Corporation, Tokyo, Japan) and calculated by using software (NIS-Elements BR4.00.12). The bubble diameter was measured twice per day for 4 consecutive days. Data were collected from at least 50 bubbles each time.

## 3. Results and Discussion

### 3.1. Foam Morphology during A. platensis Cultivation

The foam morphology above the solution was observed during algal cultivation. With increasing microalgae density in solution, the surface tension decreased, which weakened the interfacial properties at the air-liquid interface. Therefore, the microalgal solution resulted in fast adsorption kinetics and high surface visco-elastic interface properties. These properties are favorable for foam formation [15,27,28]. Figure 3 illustrates the process involved in stable and unstable bubble formation and a typical mechanism of foam formation during the cultivation of *A. platensis*. A sticky gel-like material was continuously excreted through the walls of *A. platensis,* known as extracellular polymeric substance (EPS), that generated an aggregate formation by adhesion between microalgae surfaces which were trapped in the thin bubble films that promoted stable foam over the solution surface [27,28].

As shown in Figure 4a, the foam height over the surface of microalgae solution gradually increased and reached the highest value of 30 mm after 4 days (Figure 4b). During this time, the bubble diameter also increased and reached 1.8 mm after 4 days (Figure 4b). Importantly, the bubble diameter close to the surface was smaller as compared to the bubbles located at the top of foams and away from the solution. This phenomenon occurred due to coarsening.

Bubbles generated by aeration can resist bursting because adsorption of the sticky nature of algal cells attached to bubbles led to the formation of a resistant armor that slowed down the drainage, reduced bubble breakage, and increased coarsening, which resulted in more stable and long-lasting foam [29]. As the foam height rises, bubble diameter increases, and bubbles coarsen from smaller bubbles to larger bubbles over time due to gas diffusion [30]. According to the Laplace–Young law, ‘coarsening’ involves the transport of gas between bubbles due to their differences in pressure, leading to an increase in the average bubble diameter over time [31,32].

### 3.2. Stability of Foam during Biomass Production

Foaming was a gradual and continuous process during the cultivation of microalgae, and 1.5 g/L of microalgal biomass was produced after 4 days (Figure 5). When the biomass grew denser and *A. platensis* cells started to produce EPS around their walls, a large portion of EPS was attached to the surface of *A. platensis* cells, whereas some were released into the culture medium. The separation of EPS from the cell surface is a well-established phenomenon [30,33,34]. The released EPS formed stable thin films in the solution and resulted in the formation of stable foams. This polymeric substance, i.e., EPS, consists of sulfated substitutes (0.5–22%), uronic acids (14–40%), and polysaccharides. The results showed an association between the microalgal biomass and surface tension. The surface tension continuously decreased with increasing microalgal biomass (Figure 5). The amphiphilic nature of the algal solution is due to the presence of uronic acids and peptides in EPS [35]. Therefore, the emulsifying property of the solution was significantly enhanced, which is caused by the presence of rhamnose and fucose deoxy sugars [29,33,36]. During the cultivation time, the pH of the solution also increased, although only slightly, from ~9.8 to ~10.2 (Figure 5). This slight increase in pH during foam formation could be due to the CO_2_ uptake.

The dependence of foaming characteristics, including FS, FC, FE, and FMD, on the microalgal growth rate was monitored (Figure 6). FC increased steadily with increasing dry biomass weight and reached 1.2 when 1.5 g/L dry biomass was produced after 4 days. A similar trend was found for FE that increased from 15 to 43 when dry biomass of 1.5 g/L was produced after 4 days. The foam produced by 1.5 g/L microalgal dry biomass remained stable for 10 s. Similarly, the FMD was increased from 0.02 to 0.07 during the formation of 1.5 g/L microalgal dry biomass. The results showed that all foaming characteristics, including FS, FC, FE, and FMD, were increased with the increasing microalgal biomass dry weight. Bubble interfaces were accumulated by microalgae together with EPS and formed a soft adhesive gel that favoured the formation of stable and long-lasting foams, as shown in Figure 4. As the microalgal density increased, the cells were adsorbed by the bubble and carried over to the surface of the solution during aeration or paddle wheel rotation. The EPS excretion gradually decreased the surface tension of the solution because EPS behaved as a surfactant and increased the stickiness of the culture, therefore creating a positive environment to generate stable foaming [33,37].

### 3.3. Effect of CO_2_ Aeration on Foam Stability

Since CO_2_ gas influenced the rapid volume expansion of foam when the bioreactor was continuously aerated from the bottom, the EPS attached to the *A. platensis* cell wall enhanced the formation of stable foams. The maximum solubility of CO_2_ in an algal solution was essential for efficient algal growth, but when the aeration rate became greater than the solubility capability of the solution, the supplied gas went through the algal solution idly and contributed to the foam formation on the surface of the solution.

The effect of CO_2_ flow rate on the formation of foam was investigated by using the Foamscan instrument that helped to analyze four different characteristics of the algal foam, including the FS, FC, FE, and FMD. Figure 7a shows that the FS of the solution increased gradually from 1 to 10 s during the 4 days as the algal growth increased. It also showed that FC and FE increased from 0.26 ± 0.03 to 0.90 ± 0.03 and 22 ± 0.46 to 38 ± 0.46 over time (Figure 7a,b). These results indicate that CO_2_ aeration and *A. platensis* cell density increased simultaneously. FMD was the only parameter that decreased from 5.5 ± 2.8 to 3.1 × 10^−2^ ± 2.8 with the increasing aeration rate of gas (Figure 7b). This could be attributed to the continuous supply of CO_2_ gas that led to rapid bubble formation that ruptured more quickly due to the high aeration rate. Thus, larger bubbles with lower density were formed at higher flow rates.

### 3.4. Effect of Cultivation Temperature on Foaming Stability

Temperature is one of the most important control parameters during the algal growth, as the metabolic and enzymatic activities are directly influenced by it during the simultaneous production of biomass and EPS [34,36,38]. A higher temperature reduces the solubility of gaseous components (i.e., CO_2_) in the culture medium, whereas a lower temperature leads to reduced algal growth and decreases the kinetics of metabolic activities [38].

The experimental data show that temperature greatly contributed to affecting the foaming events. A higher temperature affected the bubble drag, bubble rise velocity, surface tension, and flow behavior of the foam, which led to an increased mixing and gaseous hold up within the bubbles. These results are in accordance with a previous study [18]. The foam film permeability increased with the increasing temperature, whereas the surface tension decreased, as reported previously [39]. Temperature behaves as an indirect contributor to foaming events in photo-bioreactors. In Figure 8a,b, it is shown that foaming characteristics increased during the culture, such as FS from 0 to 9 s, FC from 0.28 ± 0.03 to 0.89 ± 0.03, and FE from 26 ± 0.41 to 38 ± 0.41. On the other hand, only FMD decreased from 6.1 ± 0.28 to 3.0 × 10^−2^ ± 0.28 (Figure 8b). The decreased FMD could be due to the fact that a higher temperature resulted in the formation of larger but less dense bubbles, which caused rapid drainage of liquid through the film-forming unstable foam.

## 4. Conclusions

The foaming mechanism in algal culture was investigated with varying CO_2_ flow rates and temperature using a Foamscan instrument. It is concluded that, as the microalgae solution became denser and reached 2.4 g/L, the adhesive EPS released from *A. platensis* decreased the surface tension to 45 mN/m. Consequently, an increased CO_2_ flow rate of 400 mL/min and a temperature of 40 °C increased the stability of the foam up to 10 s with 30 mm height and 1.8 mm bubble diameter. Based on the findings of this study, it is recommended that a high temperature and rigorous CO_2_ aeration should be avoided to mitigate the foaming problem in photobioreactors.

## Figures and Tables

**Figure 1 bioengineering-09-00257-f001:**
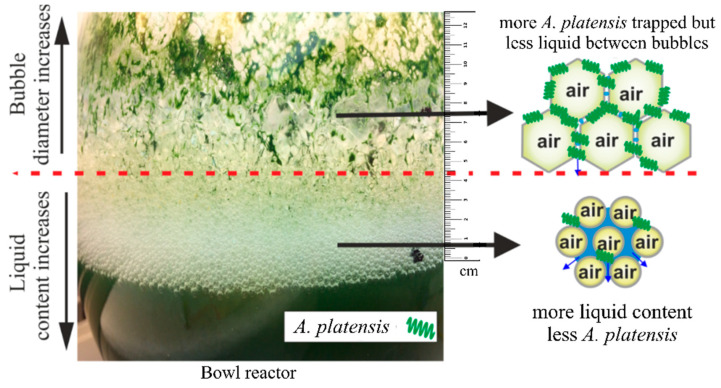
Formation of stable CO_2_ bubbles at different foam heights.

**Figure 2 bioengineering-09-00257-f002:**
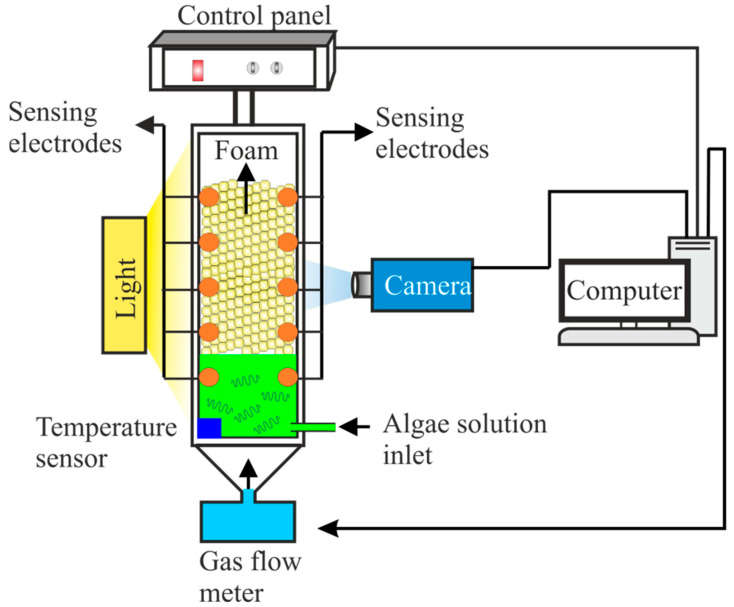
Schematic illustration of foam measurement system.

**Figure 3 bioengineering-09-00257-f003:**
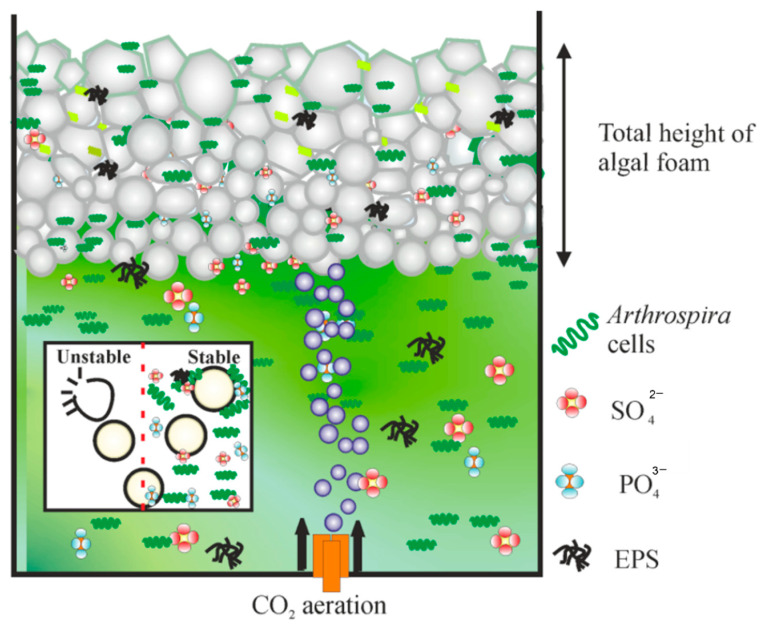
Mechanism of foam formation during cultivation of *A. platensis*.

**Figure 4 bioengineering-09-00257-f004:**
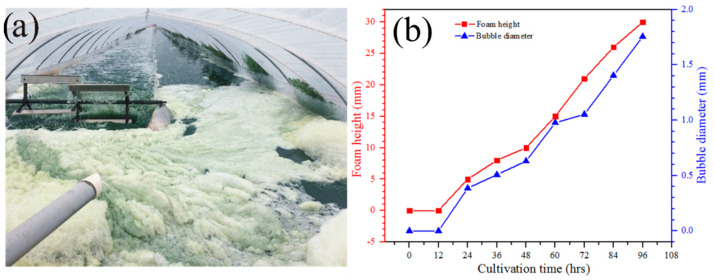
(**a**) Foam formation during cultivation of *A. platensis* cultivation in industrial raceway ponds (**b**) Qualitative and quantitative demonstration of increasing foam height and bubble diameter after 4 days of microalgal growth.

**Figure 5 bioengineering-09-00257-f005:**
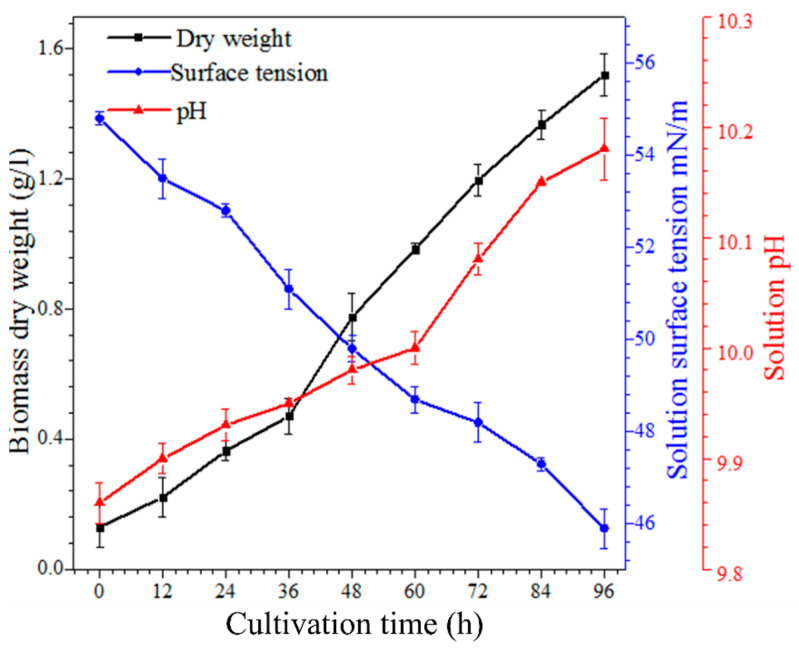
Formation of microalgal biomass and its impact on the solution surface tension and pH during the 4 days growth of *A. platensis*. The algal biomass increased with time, which increased the pH and decreased the surface tension of the solution.

**Figure 6 bioengineering-09-00257-f006:**
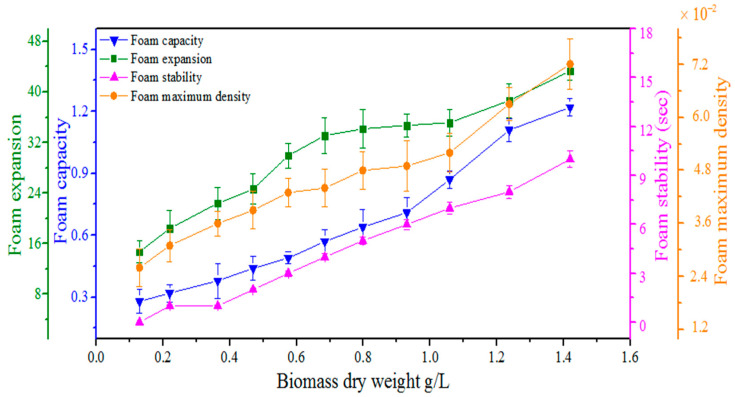
The dependence of foaming characteristics, including foam capacity, foam expansion, foam stability, and foam maximum density, on the microalgal biomass dry weight produced for 4 days. Note: **Foam Stability** = Half-life of foam from generation to disappear, **Foam capacity** = Ratio of foam volume to gas volume, **Foam expansion** = Ratio of foam volume to reduced liquid volume from initial to final condition, and **Foam maximum density** = Ratio of the difference between the initial liquid volume and the final liquid volume with the final foam volume. Note: Each data point is an average of three replicates with a standard deviation, *n* = 3.

**Figure 7 bioengineering-09-00257-f007:**
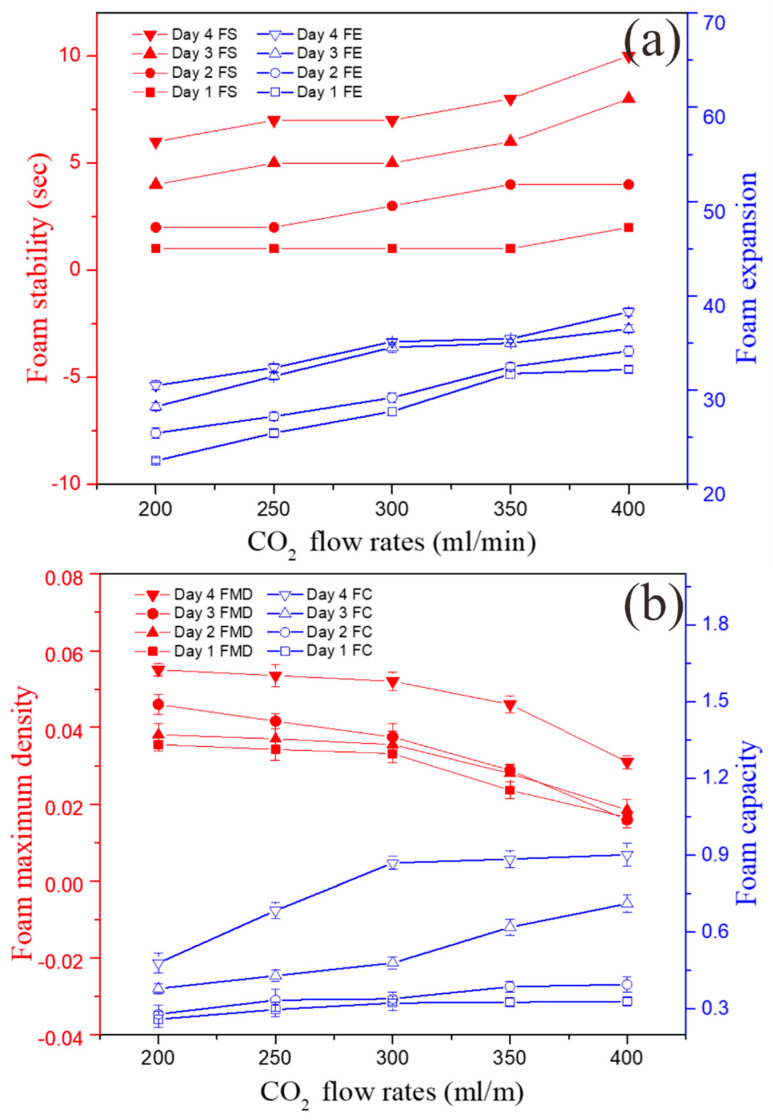
Effects of CO_2_ flow rates on dynamic characteristics: (**a**) foam stability and expansion and (**b**) foam maximum density and capacity.

**Figure 8 bioengineering-09-00257-f008:**
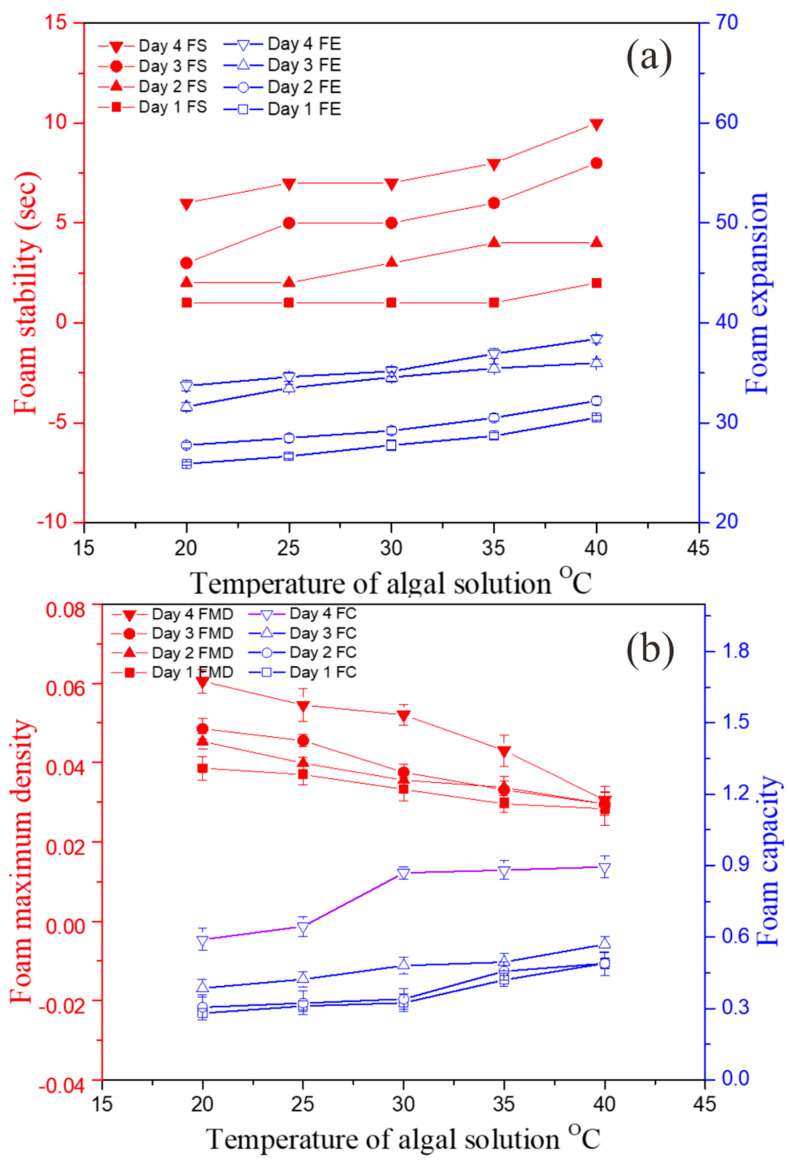
Effects of temperatures on dynamic characteristics: (**a**) foam stability and expansion and (**b**) foam maximum density and capacity.

## Data Availability

Not applicable.

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
