# Peer review of "Dynamic Foam Characteristics during Cultivation of Arthrospira platensis"

_bioengineering, 2022, doi:10.3390/bioengineering9060257_

Round 1

Reviewer 1 Report

Figure 1: Replace Arthrospira by A. platensis; include a scale
L 62: A. platensis
Line 52-67 no literature cited

L 71: Foamscan instrument is shown in Figure 2 --> The instrument is not explained in Figure 2. 

Chapter 2.2

The experimental set up is not explained satisfying. A scetch would be useful.  CO2 is not sparged into an open pond. The experimental set up does not simulate the actual conditions in a receway/open pond
Which CO2 concentration was used and why?
Chapter 2.4

What is bowl reactor?
Why do you have an air inlet? Why is it not a small raceway pond/Open pond to simulate real conditions?
How much light was added? And where are the light sources placed? Which light sources were used? Which DAy/Night Rythm was used and why?

Chapter 3.1

What is shown in Figure 3A
Why are the main results shown in Figure 2? 

Chapter 3.2
EPS formation is discussed, but how was it extracted and how was it analyzed? Where are the results?
Caption of Figure 4 is not sufficient. Cultivation parameteres are missing, Amounts of replicates are missing etc.
Where is the connection between foam formation and EPS amount and composition
Where is the connection between pH and foam formation? Increasing pH is normal in a microalgal culture due to CO2 uptake
Foam formation, densitiy etc. is not well described and discussed. Correlations can't be seen, or are not explained.

Chapter 3.3
Aeration of CO2 causes foam formation? I would say the air inlet is the problem and not a specific CO2 concentration
L180-186 is not ounderstandable
Whole set up is not well described? Different flow rates of CO2 or just air? Which concentrations were used and why? Were is the comparison to real conditions? Caption of Figure 6 is not sufficient. Which mathematical model is used to connect the data points in FIgure 6?

Chapter 3.4
EPS is not a by-product it is part of the biomass. 
L 202-209 should be in the material and method section
It does not become clear if foam formation is caused by the temperature or by an increase in biomass concentration

Reviewer 2 Report

The document presented by the authors is well organized and presents interesting research on foam formation in the production of A. platensis and its possible drawbacks.

my only comment is about the addition of figure 3. in the methods the authors describe the production of A. platensis in a "bowl-photobioreactor", but figure 3 presents a raceway system. if the authors tested the foam formation in large systems it is also important to add the culture conditions. 
